# Socioeconomic characteristics, family structure and trajectories of children's psychosocial problems in a period of social transition

Daniela Kuruczova[1]*, Jana Klanova[1], Jiri Jarkovsky[1,2], Hynek Pikhart[1,3], Julie Bienertova-Vasku[1]

1 Research Centre for Toxic Compounds in the Environment, Faculty of Science, Masaryk University, Brno, Czech Republic, 2 Institute for Biostatistics and Analyses, Masaryk University, Brno, Czech Republic, 3 Research Department of Epidemiology and Public Health, University College London, London, United Kingdom

* daniela.kuruczova@mail.muni.cz

**Data Availability Statement:** Datasets generated and/or analysed in the current study are available on reasonable request through the website of the

## Abstract

Data from the Czech part of the European Longitudinal Study of Pregnancy and Childhood offer a unique opportunity to examine a period of changing socioeconomic structure of the country. Our aim was to analyse the association between socioeconomic status, family structure and children's psychosocial problems at the age of 7, 11, 15 and 18 years in 3,261 subjects and compare our results with findings from western settings. The Strengths and Difficulties Questionnaire (SDQ) and its five subscales were used to assess individual problem areas (emotional symptoms, peer problems, hyperactivity, conduct problems) and prosocial behaviour. Socioeconomic status was represented by maternal education and three forms of family structure were identified: nuclear family, new partner family and single parent family. The SDQ subscale score over time was modelled as a quadratic growth curve using a linear mixed-effects model. Maternal university education was associated with a faster decline in problems over time for all five SDQ subscales. Problems in children from nuclear families were found to be significantly lower than in children from single parent families for all SDQ subscales with the exception of peer problems. Compared to nuclear families, children from new partner families scored significantly higher in hyperactivity and conduct problems subscales. The nuclear family structure and higher maternal education have been identified as protective factors for children's psychosocial problems, in agreement with findings from western settings. Adopting a longitudinal perspective was shown as essential for providing a more complex view of children's psychosocial problems over time.

## Introduction

The relationship between psychosocial problems in children, socio-economic status (SES), and family structure has been previously explored. Multiple studies suggest that both high SES and

Czech ELSPAC project: http://www.elspac.cz/index-en.php. The analysed dataset is not freely available for download due to concerns for subjects' identifiability.

**Funding:** The ELSPAC study is currently supported by the RECETOX research infrastructure (LM2018121). This research was supported by the Horizon 2020 Teaming 2 project (857560) and the Ministry of Education, Youth and Sports of the Czech republic (CZ.02.1.01/0.0/0.0/17_043/0009632 and CZ.02.1.01/0.0/0.0/15_003/0000469). The funders had no role in study design, data collection and analysis, decision to publish, or preparation of the manuscript.

**Competing interests:** The authors have declared that no competing interests exist.

nuclear parent family status are associated with a decrease in children's psychosocial problems. When it comes to the SES in general, other variables may moderate its mechanism of influence: low socioeconomic status is associated with higher stress, worse parenting style, and poor social environment [1]. These variables may, in turn, influence the child. In the context of individual SES components, high maternal education and high family income have been identified as protective factors in both pre-adolescent [2] and adolescent children [3–5]. Research into family structure reveals a compelling association between single parent family structure and a higher rate of psychosocial problems in children [2,4–7]. The role of the new-partner family, where the parent entered a new relationship or re-married, is not so well-researched. Some studies focus purely on single parent families and pool all other family types together [5,6]. However, studies which do include this family type suggest that its effect is not straightforward; multiple moderators influencing the association, such as SES or relationships within the reconstituted family, have been proposed [4,8,9].

It is worth noting that the majority of research focusing on psychosocial problems in children is cross-sectional. While some studies have found absolute differences and associations at specific time-points, adding information from other time-points may help reveal more complex relationships and interactions between variables. When it comes to the psychosocial problems score, the cross-sectional approach is capable of revealing an absolute difference associated with a given variable. However, it cannot detect a situation where another variable is associated with a change of the progression of the problem score, e.g. increase or decrease over time. This benefit of using a longitudinal approach is exemplified in a study by Flouri, Midouhas, & Ruddy [5], which modelled the children's problem trajectories as a quadratic growth curve. Their results suggest that family structure and SES not only influence the absolute difference in the score but also, in some cases, the specific progression of the problem score over time. Another notable longitudinal perspective focuses on adding more information to the family structure variable by focusing on the quantity and quality of family structure transitions. For example, a study by Ryan and Classens [9] explores the effect of change in family structure on children's behaviour problems in time. Results suggest that the number of transitions itself has a negative impact on the children's psychosocial problems, pointing out the transition to the single parent family as the most problematic one [9,10].

The vast majority of research presented above was conducted on samples from the United Kingdom, United States, Australia, or other developed countries. A search query on a website dedicated to SDQ [11,12], a screening questionnaire for children's psychosocial problems translated into over 80 languages, revealed over 2300 studies from the United Kingdom, United States, Canada, or Australia. On the other hand, a query on Eastern European post-communist countries such as Czech Republic, Hungary, Poland, or Slovakia returned less than 50 results. The most notable research project examining children's psychosocial problems, which included Czech, Hungarian and Polish populations, was the European KIDSCREEN Study conducted in the 2000s [13]. Of these three countries, the prevalence of psychosocial problems was found to be highest in the Czech Republic, with 13.3% borderline and 7.4% abnormal cases, but still behind the UK (13.2% and 10.4%). This study also suggests that poor social support, parental relation, and parental mental health are associated with worse psychosocial problems in Czech and Polish children. In contrast to the UK sample, no relationship between psychosocial problems and SES was found in either the Czech Republic or in Poland. The other notable finding of this study reveals the difference in SES; the proportion of children living in families with high SES is much lower in Eastern European countries compared to European countries with no history of communist regimes.

The distinguishing event in the history of Eastern European countries constitutes a major socioeconomic transition. Since the fall of the communist regime in 1989, post-communist

countries have undergone a transformation from a command economy to a market-oriented economy. The period between the 1990s and 2000s brought about rapid economic and social change in the Czech Republic. The initial transitional recession was followed by economic growth and an entrepreneurial boom [14,15]. Income inequality, which was considered low at the beginning of the transition, began to rise [16,17]. Likewise, the divorce rate grew gradually, and the proportion of single parent or reconstituted families increased [18,19].

Data from the Czech part of the European Longitudinal Study of Pregnancy and Childhood (ELSPAC) provide us with a unique opportunity to study this period from a longitudinal perspective. Our aim is to study the association between SES, family structure, and psychosocial problems in children over time and compare our results with findings from the western settings. We anticipate that the mechanisms already described in existing literature are robust and applicable for this specific time period. We therefore expect our findings to comply with these mechanisms, especially with respect to apparent risk factors such as low SES or single parent families. We also expect the effect size to be less pronounced due to several reasons. First, the surveyed period was a period of changes, including (among other things) a rise in income inequality and divorce rate. Second, the results from the KIDSCREEN Study [13] suggest that risk factors for psychosocial problems have somewhat lower odds in the Czech Republic, especially in comparison with the UK. We believe that our study can test previously established findings in a somewhat different setting while adding to existing research results thanks to the use of a longitudinal approach.

## Materials and methods

### Study population

The European Longitudinal Study of Pregnancy and Childhood (ELSPAC) [20] was initiated by the World Health Organisation in 1985. The study was designed to investigate the effects of various biological, environmental, social, economic, and psychological factors on a child's health from the mother's pregnancy to the child's adult age. The study design was coordinated with other European longitudinal studies from the same period (e.g., Avon Longitudinal Study of Pregnancy and Childhood [21]). A total of 5,151 children from the South Moravian region born in 1991 and 1992 were enrolled in the Czech part of the ELSPAC study.

Analysed data was collected at pre-specified ages: 7, 11, 15, and 18 (19). For this study, we used data on children's psychosocial problems only from maternal questionnaires. The choice to use only the maternal point of view was motivated by our desire to include the longest possible period of a child's life. Each subject was included in the study population if he or she had at least one time-point with complete data on at least one SDQ subscale. In total, 3,261 subjects fulfilled these conditions and were included in the analysed study population.

Ethical approval for the study was obtained from the ELSPAC Law and Ethics Committee and local research ethics committees. Written informed consent was obtained from all study participants and archived.

### Family structure

Family structure was assessed at all of four selected time-points and three mutually exclusive categories were identified: nuclear family, new partner family, and single parent family. To fall into the nuclear family category, the child had to be living with both biological parents. A family where a child was living with a biological mother and her partner who was not the child's biological father was considered a new partner family. Finally, a family where the mother lived without a partner (or did not have one) was considered a single parent family. Due to limited data on children not living with their biological mothers, family structure was assessed only

from the mother's point of view. All other family structures (e.g. families with single fathers) were scarce in the dataset and therefore excluded. Family structure data was not collected at 18y, but rather at 19y. Since changes in family structure during this interval may be considered negligible, family structure at 19y was used for the 18y time-point.

## Socioeconomic status

SES was represented only by one variable–maternal education level at the time of pregnancy. This choice is supported by several arguments. First, as the focus of this study is family structure, using data on biological father might have had an unpredictable effect for single parent and new partner families. Second, additional socioeconomic variables such as maternal employment or family income are known to correlate strongly with education level. Finally, the selected variable had a considerably higher response rate than information on family income.

## Psychosocial problems in children

The Czech version of the Strengths and Difficulties Questionnaire (SDQ) [11] was used to assess children's problems. The SDQ consists of five subscales, four of them focusing on problem areas: emotional symptoms, conduct problems, hyperactivity, and peer problems. The emotional symptoms and peer problems can be grouped as internalising subscales, expressing internal psychological problems of the child, while conduct problems and hyperactivity subscales are externalising subscales with problems usually manifesting in a child's behaviour. The fifth subscale measures the child's prosocial behaviour. All items are rated on a three-point scale from "not true" to "somewhat true" to "certainly true" and each subscale consists of 5 items. The ratings are subsequently added up to create subscale scores ranging from 0 to 10. As per official scoring recommendations [12], the subscale score is considered valid if 3 or more items out of 5 have been answered. In the case of missing answers, the mean score is calculated and multiplied by 5. The questionnaire may be completed by a parent, teacher, or, from a certain age, by the child. In our study, it was filled out by mothers at 7, 11, 15, and 18y.

Several issues that may have affected the data quality from SDQ were identified. The translation of the questionnaire changed slightly at age 15, but the meaning of individual items remained the same. Also, the questionnaire at age 11 was rated on a four-point scale and had to be converted to the original three-point version.

Despite these issues, the psychometric properties of the SDQ questionnaire in the ELSPAC sample indicate satisfactory internal consistency. The Cronbach's alpha for overall score varied over time-points and respondents in range 0.77–0.85. The internal consistency was slightly lower for all individual subscales; the hyperactivity subscale was the most consistent with alpha 0.68–0.80, followed by prosocial behaviour 0.59–0.78 and emotional symptoms 0.62–0.68. The internal consistency of the remaining two subscales was slightly lower, 0.55–0.61 for conduct problems, and 0.47–0.60 for peer problems.

## Statistical analysis

Statistical analysis was performed in the R software [22] using package nlme for model calculation [23]. First, the descriptive characteristics of the study population and basic relationships between individual variables were explored. Spearman's rank correlation coefficient [24] was used to describe relationships between individual subscales and time-points. To assess the reliability of individual SDQ subscales, Cronbach's alpha coefficient [25] was calculated.

Subsequently, we fitted a linear mixed-effects model for each subscale–a method suited for repeated measurements. This approach is especially suitable for longitudinal data as it can also

utilize data from subjects with missing data at some of the time-points, and no imputation method is thus needed [26]. The fixed effect, SDQ subscale score over time, was modelled using a quadratic polynomial growth curve. The individual changes between subjects were modelled using random intercept and slope. The mixed-effects model (without any covariates) for $Y_{ti}$–the score for $i$-th subject at the age of $t$ can be expressed as:

$$Y_{ti} = \beta_{Intercept} + \beta_{Age}t + \beta_{Age^2}t^2 + b_{0i}t + b_{1i}t^2 + \epsilon.$$

It is evident that this model is an extension of simple quadratic regression. Beta coefficients represent fixed effects which describe the entire sample, while b coefficients represent the random effects for a specific subject. The expected value of the random effect is zero; therefore, the expected value of the score at age $t$ can be expressed using only fixed effects:

$$E[Y_t] = \beta_{Intercept} + \beta_{Age}t + \beta_{Age^2}t^2$$

For each of the five SDQ subscales, several growth curve models were constructed. The variable age was centred (the mean age was subtracted from each measurement) to achieve better estimates [27]. The value of the $\beta_{Intercept}$ coefficient moves the quadratic cure along the y-axis. The additional two coefficients control the shape of the quadratic curve. If the $\beta_{Age^2}$ coefficient is zero, then the curve becomes a simple line with a slope controlled by the $\beta_{Age}$ coefficient. If it has a non-zero value, $\beta_{Age^2}$ controls the shape of the curve; for positive values, the curve has a u-shape. For negative values of $\beta_{Age^2}$ is the u-shape reversed. The actual interpretation of the shape is rather difficult using only coefficient values; a visualization of the curve is thus preferred.

Model 1 refers to the simple model without any covariates, as described above. In Model 2, the variable family structure was added, along with its interactions with age and its square. The reference level for the family structure was set to the nuclear family and dummy variables $D_{SP}$ (single parent family) and $D_{NP}$ (new partner family) were subsequently added. The formula for the expected value of the score becomes:

$$E[Y_t] = \beta_{Intercept} + \beta_{SP}D_{SP} + \beta_{NP}D_{NP} + (\beta_{Age} + \beta_{Age\times SP}D_{SP} + \beta_{Age\times NP}D_{NP})t+$$
$$(\beta_{Age^2}+\beta_{Age^2\times SP}D_{SP} + \beta_{Age^2\times NP}D_{NP})t^2.$$

Coefficients $\beta_{Intercept}$, $\beta_{Age}$ and $\beta_{Age^2}$ describe the curve for the reference level, i.e. the nuclear family. The set of coefficients for the single parent family represents the difference between the nuclear family curve and the single parent family curve. Similarly, the difference between the nuclear family curve and the new parent family curve is expressed by the new parent family coefficients.

Model 3 extends the previous model by adding two variables: the sex of the child and maternal education. Again, both variables were set to interact with both age and its square. The reference level was set to a male from a nuclear family with a mother with elementary education. The formula for the expected value is analogous to the previous one, but more dummy variables with corresponding coefficients are added. Finally, Model 4 was constructed to explore interactions between sex, maternal education, and family structure.

## Results

### Sample characteristics

The distribution of the study population over time for different variables is shown in Table 1. The proportion of males and females at all time-points is balanced and stable. Most mothers

**Table 1. Overview of sample characteristics for individual time-points and variables in the analytic and non-analytic dataset.**

| Variable | | Analytic dataset | | | Non-analytic dataset | | | p-value[*] |
|---|---|---|---|---|---|---|---|---|
| | | N | % | % nmiss | N | % | % nmiss | |
| **Sex of the child** | Male | 1,681 | 51.55 | 51.55 | 974 | 51.53 | 51.62 | 0.9857 |
| | Female | 1,580 | 48.45 | 48.45 | 913 | 48.31 | 48.38 | |
| | Missing | 0 | 0.00 | | 3 | 0.16 | | |
| **Maternal education preterm** | Elementary | 925 | 28.37 | 35.63 | 716 | 37.88 | 54.20 | <0.0001 |
| | Secondary | 1,142 | 35.02 | 43.99 | 446 | 23.60 | 33.76 | |
| | University | 529 | 16.22 | 20.38 | 159 | 8.41 | 12.04 | |
| | Missing | 665 | 20.39 | | 569 | 30.11 | | |
| **Family structure at birth** | Single parent family | 175 | 5.37 | 6.72 | 100 | 5.29 | 7.60 | 0.4651 |
| **(year 1991/92)** | New partner family | 5 | 0.15 | 0.19 | 4 | 0.21 | 0.30 | |
| | Nuclear family | 2,426 | 74.39 | 93.09 | 1,213 | 64.18 | 92.10 | |
| | Missing | 655 | 20.09 | | 573 | 30.32 | | |
| **Family structure at 7y** | Single parent family | 297 | 9.11 | 9.66 | | | | |
| **(year 1998/99)** | New partner family | 204 | 6.26 | 6.64 | | | | |
| | Nuclear family | 2,573 | 78.90 | 83.70 | | | | |
| | Missing | 187 | 5.73 | | | | | |
| **Family structure at 11y** | Single parent family | 288 | 8.83 | 11.85 | | | | |
| **(year 2002/03)** | New partner family | 218 | 6.69 | 8.97 | | | | |
| | Nuclear family | 1,925 | 59.03 | 79.18 | | | | |
| | Missing | 830 | 25.45 | | | | | |
| **Family structure at 15y** | Single parent family | 225 | 6.90 | 14.21 | | | | |
| **(year 2006/07)** | New partner family | 201 | 6.16 | 12.70 | | | | |
| | Nuclear family | 1,157 | 35.48 | 73.09 | | | | |
| | Missing | 1,678 | 51.46 | | | | | |
| **Family structure at 19y** | Single parent family | 156 | 4.78 | 16.94 | | | | |
| **(year 2010/11)** | New partner family | 132 | 4.05 | 14.33 | | | | |
| | Nuclear family | 633 | 19.14 | 68.73 | | | | |
| | Missing | 2,340 | 71.76 | | | | | |

[*]Pearson's $\chi^2$ test; % nmiss = % non-missing; NA = non-applicable

completed secondary education, followed by primary education. The most common family structure was a nuclear family at all time-points. The proportion of nuclear families, however, decreased with the increasing age of the children while the relative percentage of single parent families and new partner families rose over time. A drop-out effect typical of longitudinal studies is present, with the number of responses decreases with increasing subject age; at the final time-point, less than 50% of subjects were retained.

Table 1 also includes a comparison of the characteristics of the analytic versus non-analytic sample, i.e. subjects included in the analysis and subjects that were excluded from the analysis. In comparison with subjects excluded from the analysis, our analytic sample is biased towards better educated mothers. Family structure distribution appears to be similar in both analytic and non-analytic samples at the time of birth. Unfortunately, information on the non-analytic sample is limited from this point onward.

## Strengths and difficulties in children

Mean scores for all SDQ subscales by time-point are shown in Table 2. The mean score for all four problem subscales decreases over time, while the mean prosocial behaviour score

**Table 2. Overview of scores for individual SDQ subscales.**

| | 7y | | | | 11y | | | | 15y | | | | 18y | | | |
|---|---|---|---|---|---|---|---|---|---|---|---|---|---|---|---|---|
| | N | Nmiss (%) | Mean | SE | N | Nmiss (%) | Mean | SE | N | Nmiss (%) | Mean | SE | N | Nmiss (%) | Mean | SE |
| Emotion | 3,038 | 223 (6.84) | 1.92 | 0.032 | 2,413 | 848 (26.00) | 1.95 | 0.031 | 1,600 | 1,661 (50.94) | 1.86 | 0.044 | 1,279 | 1,982 (60.78) | 1.83 | 0.050 |
| Conduct | 3,051 | 210 (6.44) | 1.88 | 0.028 | 2,414 | 847 (25.97) | 1.70 | 0.025 | 1,600 | 1,661 (50.94) | 1.57 | 0.036 | 1,279 | 1,982 (60.78) | 1.36 | 0.038 |
| Hyperactivity | 3,037 | 224 (6.87) | 3.59 | 0.040 | 2,411 | 850 (26.00) | 3.40 | 0.044 | 1,587 | 1,674 (51.33) | 2.57 | 0.048 | 1,281 | 1,980 (60.72) | 2.28 | 0.052 |
| Peer | 3,008 | 253 (7.76) | 1.79 | 0.028 | 2,382 | 879 (26.95) | 2.04 | 0.029 | 1,590 | 1,671 (51.24) | 1.32 | 0.039 | 1,281 | 1,980 (60.72) | 1.18 | 0.041 |
| Prosocial | 3,028 | 233 (7.15) | 7.77 | 0.031 | 2,399 | 862 (26.43) | 6.72 | 0.032 | 1,598 | 1,663 (51.00) | 7.09 | 0.055 | 1,279 | 1,982 (60.78) | 7.27 | 0.059 |

N = number of subjects with valid data; Nmiss = number of subjects with missing data; SE = standard error

fluctuates between 6 and 8 points out of 10. The drop-out effect is present and most pronounced at the first three time-points, where the percentage of missing answers increases by 20% or more.

Correlations between subscales and over time (Table 3) show a stable relationship among subscales at individual time-points. It is also worth noting that correlations between the same subscales over time weaken when the time-points become more distant.

## Models

The dependence of the SDQ subscale score on age was modelled as a quadratic polynomial, allowing each variable to influence the linear as well as the quadratic coefficient of the curve. The individual results for the three growth curve models for each subscale can be found in Table 4.

In Model 1, the relationship between age and score is linear for emotional and conduct problems and quadratic for the remaining three problem subscales. All problem curves, except for peer problems, decrease over time. The peer problems score increases until approximately 10y and then begins to decrease. The prosocial behaviour score has a pronounced u-shape.

Model 2 introduces family structure with the nuclear family as the reference level. The reference level curves for the nuclear family are similar to those from Model 1. Children from single parent families have a significantly worse score in all problem subscales with the exception of peer problems. The prosocial behaviour score curve for children from single parent families has a significantly different linear coefficient and subsequently less pronounced u-shape. Children from new partner families exhibit significantly worse results with respect to the conduct problems subscale and have a significantly different quadratic coefficient in the prosocial behaviour scale, resulting in a less distinct u-shape. A significant difference in the linear coefficient is present for emotional symptoms, leading to a gradual decrease in the problem score over time.

Growth curves constructed in accordance with Model 3 are shown in Fig 1. The introduction of the variable sex revealed significant differences between the scores achieved by male and female subjects in all subscales, with females achieving a significantly lower problem score and a higher prosocial behaviour score. The difference is mostly expressed as a simple vertical shift with the notable exception of the emotional symptoms subscale, where the shape of the curve depends on the sex of the child–the score decreases over time for boys and increases over time for girls. The shape of the curve is also different in the hyperactivity subscale, where the girl's curve seems linear and decreasing, while the boy's curve is a quadratic polynomial. Maternal education is significant for all subscales, where higher education contributed to a lower score or a more steeply decreasing curve. This trend is visible in the curve shape for

**Table 3. Spearman rank correlations between SDQ subscales over time.**

| | | 7y | | | | | 11y | | | | | 15y | | | | | 18y | | | |
|---|---|---|---|---|---|---|---|---|---|---|---|---|---|---|---|---|---|---|---|---|
| | | E | C | H | Pea | Pr | E | C | H | Pe | Pr | E | C | H | Pe | Pr | E | C | H | Pe |
| 7y | Conduct | 0.25 | | | | | | | | | | | | | | | | | | |
| | Hyper | 0.29 | 0.50 | | | | | | | | | | | | | | | | | |
| | Peer | 0.33 | 0.23 | 0.22 | | | | | | | | | | | | | | | | |
| | Prosocial | -0.01[NS] | -0.33 | -0.25 | -0.16 | | | | | | | | | | | | | | | |
| 11y | Emotion | 0.44 | 0.15 | 0.21 | 0.21 | -0.07 | | | | | | | | | | | | | | |
| | Conduct | 0.14 | 0.46 | 0.36 | 0.17 | -0.25 | 0.25 | | | | | | | | | | | | | |
| | Hyper | 0.12 | 0.38 | 0.56 | 0.12 | -0.19 | 0.26 | 0.49 | | | | | | | | | | | | |
| | Peer | 0.19 | 0.14 | 0.19 | 0.36 | -0.12 | 0.35 | 0.26 | 0.19 | | | | | | | | | | | |
| | Prosocial | -0.04[NS] | -0.22 | -0.19 | -0.13 | 0.40 | -0.04 | -0.33 | -0.23 | -0.20 | | | | | | | | | | |
| 15y | Emotion | 0.28 | 0.11 | 0.13 | 0.11 | 0.05 | 0.39 | 0.12 | 0.14 | 0.19 | 0.03[NS] | | | | | | | | | |
| | Conduct | 0.15 | 0.35 | 0.27 | 0.09 | -0.13 | 0.19 | 0.42 | 0.30 | 0.08 | -0.18 | 0.28 | | | | | | | | |
| | Hyper | 0.16 | 0.28 | 0.39 | 0.13 | -0.15 | 0.21 | 0.33 | 0.49 | 0.14 | -0.14 | 0.33 | 0.53 | | | | | | | |
| | Peer | 0.15 | 0.10 | 0.12 | 0.25 | -0.12 | 0.20 | 0.10 | 0.04[NS] | 0.36 | -0.12 | 0.21 | 0.08 | 0.13 | | | | | | |
| | Prosocial | -0.04[NS] | -0.25 | -0.21 | -0.10 | 0.34 | -0.08 | -0.26 | -0.19 | -0.09 | 0.42 | 0.07 | -0.38 | -0.31 | -0.12 | | | | | |
| 18y | Emotion | 0.26 | 0.10 | 0.15 | 0.14 | 0.04[NS] | 0.31 | 0.15 | 0.17 | 0.18 | 0.03[NS] | 0.46 | 0.22 | 0.22 | 0.13 | 0.01[NS] | | | | |
| | Conduct | 0.09 | 0.31 | 0.27 | 0.10 | -0.17 | 0.16 | 0.34 | 0.30 | 0.07 | -0.20 | 0.20 | 0.52 | 0.37 | 0.06 | -0.28 | 0.30 | | | |
| | Hyper | 0.16 | 0.27 | 0.35 | 0.12 | -0.16 | 0.22 | 0.32 | 0.41 | 0.15 | -0.18 | 0.21 | 0.36 | 0.52 | 0.08 | -0.26 | 0.39 | 0.58 | | |
| | Peer | 0.15 | 0.10 | 0.13 | 0.25 | -0.14 | 0.19 | 0.14 | 0.08 | 0.32 | -0.16 | 0.16 | 0.11 | 0.10 | 0.45 | -0.12 | 0.31 | 0.17 | 0.20 | |
| | Prosocial | -0.06 | -0.21 | -0.15 | -0.12 | 0.32 | -0.09 | -0.22 | -0.13 | -0.06 | 0.38 | 0.04[NS] | -0.25 | -0.20 | -0.13 | 0.57 | -0.02[NS] | -0.42 | -0.35 | -0.23 |

All correlation coefficients are significant with p < 0.05 unless specified otherwise; E = Emotion, C = Conduct, H = Hyperactivity, Pe = Peer, Pr = Prosocial; [NS] = p > 0.05

different education levels in almost all problem subscales, with the most notable change in the case of the emotional symptoms subscale (Fig 1, first row). The higher the maternal education, the steeper the decrease, i.e. problems score for children of mothers with higher education decreased faster over time. The absolute difference is most pronounced in the hyperactivity subscale, where maternal university education is tied to a significantly lower score. Maternal university education is also associated with a lower score on the prosocial behaviour subscale. The majority of associations with family structure from Model 2 were retained, with minor changes in coefficient values.

Interactions between individual variables were explored as well. However, as the results remain largely the same, and since very few significant interactions were identified, the full results are not included. The only notable significant interaction was found in case of hyperactivity and conduct subscales for a combination of high school education, new partner family, and quadratic coefficient.

## Discussion

We aimed to explore the relationship between children's problems and family structure at a time of socioeconomic change in the Czech Republic. The children included in this study were born several years after the fall of the communist regime and grew up in a period of transition towards capitalism.

Studies from western settings have previously shown an association between children's psychosocial problems and family structure [2,5]. Specifically, single parenthood has been shown to result in an increased risk of psychological and financial burdens and has been associated

**Table 4. Results of growth curve models.**

| SUBSCALE | VARIABLE | MODEL 1 | p-value | MODEL 2 | p-value | MODEL 3 | p-value |
|---|---|---|---|---|---|---|---|
| Emotional symptoms | Intercept | 1.92(0.031) | <0.001 | 1.87(0.035) | <0.001 | 1.73(0.073) | <0.001 |
| | Age | -0.01(0.005) | 0.039 | -0.02(0.006) | <0.001 | -0.05(0.014) | 0.001 |
| | Age$^2$ | 0.00(0.001) | 0.106 | 0.00(0.001) | 0.018 | 0.00(0.003) | 0.524 |
| | New partner family | | | 0.03(0.097) | 0.772 | 0.01(0.111) | 0.920 |
| | New partner family x age | | | 0.04(0.017) | 0.043 | 0.02(0.019) | 0.327 |
| | New partner family x age$^2$ | | | 0.01(0.004) | 0.212 | 0.00(0.005) | 0.344 |
| | Single parent | | | 0.37(0.088) | <0.001 | 0.36(0.099) | <0.001 |
| | Single parent x age | | | 0.02(0.016) | 0.239 | 0.01(0.018) | 0.746 |
| | Single parent x age$^2$ | | | 0.00(0.004) | 0.999 | 0.00(0.004) | 0.325 |
| | Female | | | | | 0.27(0.071) | <0.001 |
| | Female x age | | | | | 0.07(0.012) | <0.001 |
| | Female x age$^2$ | | | | | 0.00(0.003) | 0.099 |
| | High school | | | | | 0.01(0.081) | 0.873 |
| | High school x age | | | | | 0.00(0.015) | 0.914 |
| | High school x age$^2$ | | | | | 0.00(0.003) | 0.143 |
| | University | | | | | 0.06(0.098) | 0.568 |
| | University x age | | | | | -0.03(0.017) | 0.131 |
| | University x age$^2$ | | | | | -0.01(0.004) | 0.018 |
| Conduct problems | Intercept | 1.66(0.025) | <0.001 | 1.60(0.029) | <0.001 | 1.72(0.059) | <0.001 |
| | Age | -0.04(0.004) | <0.001 | -0.05(0.005) | <0.001 | -0.05(0.011) | <0.001 |
| | Age$^2$ | 0.00(0.001) | 0.428 | 0.00(0.001) | 0.328 | 0.00(0.002) | 0.074 |
| | New partner family | | | 0.32(0.078) | <0.001 | 0.40(0.089) | <0.001 |
| | New partner family x age | | | 0.00(0.014) | 0.758 | -0.01(0.016) | 0.395 |
| | New partner family x age$^2$ | | | 0.00(0.003) | 0.261 | -0.01(0.004) | 0.165 |
| | Single parent | | | 0.18(0.071) | 0.011 | 0.23(0.080) | 0.004 |
| | Single parent x age | | | 0.01(0.013) | 0.346 | 0.01(0.014) | 0.534 |
| | Single parent x age$^2$ | | | 0.00(0.003) | 0.514 | 0.00(0.003) | 0.979 |
| | Female | | | | | -0.26(0.057) | <0.001 |
| | Female x age | | | | | 0.02(0.010) | 0.047 |
| | Female x age$^2$ | | | | | 0.00(0.002) | 0.078 |
| | High school | | | | | -0.02(0.066) | 0.812 |
| | High school x age | | | | | 0.01(0.012) | 0.563 |
| | High school x age$^2$ | | | | | 0.00(0.003) | 0.119 |
| | University | | | | | 0.00(0.080) | 0.952 |
| | University x age | | | | | -0.04(0.014) | 0.007 |
| | University x age$^2$ | | | | | -0.01(0.003) | 0.046 |

(*Continued*)

**Table 4.** (Continued)

| SUBSCALE | VARIABLE | MODEL 1 | p-value | MODEL 2 | p-value | MODEL 3 | p-value |
|---|---|---|---|---|---|---|---|
| Hyperactivity | Intercept | 3.10(0.037) | <0.001 | 3.02(0.042) | <0.001 | 3.53(0.087) | <0.001 |
| | Age | -0.13(0.005) | <0.001 | -0.14(0.007) | <0.001 | -0.15(0.015) | <0.001 |
| | Age$^2$ | -0.01(0.001) | <0.001 | -0.01(0.002) | <0.001 | -0.02(0.003) | <0.001 |
| | New partner family | | | 0.20(0.114) | 0.075 | 0.26(0.130) | 0.043 |
| | New partner family x age | | | 0.01(0.020) | 0.536 | 0.01(0.022) | 0.691 |
| | New partner family x age$^2$ | | | 0.01(0.005) | 0.133 | 0.01(0.006) | 0.340 |
| | Single parent | | | 0.35(0.104) | 0.001 | 0.37(0.116) | 0.001 |
| | Single parent x age | | | 0.01(0.017) | 0.764 | -0.01(0.020) | 0.607 |
| | Single parent x age$^2$ | | | 0.00(0.004) | 0.912 | 0.00(0.005) | 0.992 |
| | Female | | | | | -0.77(0.085) | <0.001 |
| | Female x age | | | | | 0.05(0.013) | 0.001 |
| | Female x age$^2$ | | | | | 0.01(0.003) | <0.001 |
| | High school | | | | | -0.13(0.097) | 0.194 |
| | High school x age | | | | | -0.01(0.016) | 0.412 |
| | High school x age$^2$ | | | | | 0.00(0.004) | 0.564 |
| | University | | | | | -0.28(0.118) | 0.019 |
| | University x age | | | | | -0.04(0.018) | 0.032 |
| | University x age$^2$ | | | | | 0.00(0.004) | 0.687 |
| Peer problems | Intercept | 1.82(0.027) | <0.001 | 1.78(0.032) | <0.001 | 1.91(0.065) | <0.001 |
| | Age | -0.07(0.004) | <0.001 | -0.09(0.006) | <0.001 | -0.06(0.012) | <0.001 |
| | Age$^2$ | -0.01(0.001) | <0.001 | -0.01(0.001) | <0.001 | -0.01(0.003) | <0.001 |
| | New partner family | | | 0.08(0.089) | 0.355 | 0.10(0.101) | 0.303 |
| | New partner family x age | | | -0.01(0.015) | 0.705 | -0.01(0.017) | 0.556 |
| | New partner family x age$^2$ | | | 0.00(0.004) | 0.756 | 0.00(0.004) | 0.952 |
| | Single parent | | | 0.13(0.081) | 0.104 | 0.12(0.090) | 0.189 |
| | Single parent x age | | | 0.01(0.014) | 0.379 | 0.01(0.016) | 0.623 |
| | Single parent x age$^2$ | | | 0.00(0.004) | 0.682 | 0.00(0.004) | 0.763 |
| | Female | | | | | -0.17(0.063) | 0.007 |
| | Female x age | | | | | -0.01(0.011) | 0.226 |
| | Female x age$^2$ | | | | | 0.00(0.003) | 0.073 |
| | High school | | | | | -0.12(0.073) | 0.098 |
| | High school x age | | | | | -0.01(0.013) | 0.311 |
| | High school x age$^2$ | | | | | 0.00(0.003) | 0.291 |
| | University | | | | | 0.03(0.087) | 0.709 |
| | University x age | | | | | -0.04(0.015) | 0.004 |
| | University x age$^2$ | | | | | -0.01(0.004) | 0.074 |

(*Continued*)

**Table 4.** (Continued)

| SUBSCALE | VARIABLE | MODEL 1 | p-value | MODEL 2 | p-value | MODEL 3 | p-value |
|---|---|---|---|---|---|---|---|
| Prosocial behaviour | Intercept | 6.75(0.035) | <0.001 | 6.73(0.040) | <0.001 | 6.51(0.083) | <0.001 |
| | Age | -0.02(0.005) | <0.001 | 0.00(0.007) | 0.659 | -0.02(0.015) | 0.264 |
| | Age$^2$ | 0.03(0.001) | <0.001 | 0.03(0.002) | <0.001 | 0.03(0.003) | <0.001 |
| | New partner family | | | 0.08(0.109) | 0.459 | -0.02(0.123) | 0.865 |
| | New partner family x age | | | -0.03(0.019) | 0.096 | -0.04(0.021) | 0.053 |
| | New partner family x age$^2$ | | | -0.01(0.005) | 0.032 | -0.01(0.005) | 0.008 |
| | Single parent | | | 0.13(0.099) | 0.179 | 0.12(0.110) | 0.284 |
| | Single parent x age | | | -0.04(0.017) | 0.032 | -0.03(0.019) | 0.082 |
| | Single parent x age$^2$ | | | -0.01(0.004) | 0.076 | -0.01(0.005) | 0.054 |
| | Female | | | | | 0.67(0.080) | <0.001 |
| | Female x age | | | | | 0.03(0.014) | 0.066 |
| | Female x age$^2$ | | | | | 0.00(0.003) | 0.954 |
| | High school | | | | | -0.07(0.092) | 0.473 |
| | High school x age | | | | | 0.00(0.016) | 0.942 |
| | High school x age$^2$ | | | | | 0.00(0.003) | 0.215 |
| | University | | | | | -0.29(0.111) | 0.009 |
| | University x age | | | | | 0.02(0.019) | 0.255 |
| | University x age$^2$ | | | | | 0.01(0.004) | 0.216 |

x signifies an interaction between two variables; all random effects were significant with p<0.05.

with higher problem scores [4,6]. Our results are in agreement with these findings; the score in all SDQ problem subscales with the exception of peer problems was found to be significantly higher for children from single parent families. While new partner families consisting of two parents remove some of the burdens associated with single parent households, they may also add unpredictable relationship tensions in the family. Compared with the effects of single parent family, the effect of new partner family seems less straightforward in our results. The negative effects of the new partner family structure on the problem score were found only in case of externalising subscales (conduct and hyperactivity). In previous research, externalizing problems were associated with the quality of children's relationships with the fathers [8], which may play a role in explaining this phenomenon.

The association between higher socioeconomic status and lower psychosocial problems score found in western settings [3–5] was also confirmed in our study. The maternal education level, representing higher SES, was significant for most subscales, though the manner of influence varied. For the hyperactivity subscale, maternal university education most frequently comprised a significant negative vertical shift, i.e. the shape of the curve was the same for all education levels, while the children of university-educated mothers had lower problem scores at all ages. For all other problem subscales, the effect of maternal university education was manifested through a steeper drop of the curve over time. One possible explanation is that highly educated mothers may be better at recognising children's problems and finding suitable solutions, such as consulting specialists, which leads to a decrease of the problem score over time.

An unexpected finding is the lower prosocial behaviour subscale score in children of university-educated mothers. While the prosocial behaviour subscale is often omitted in studies using SDQ, the effect was at least expected to be in the opposite direction, i.e. higher socioeconomic status was expected to constitute a protective factor of prosocial behaviour. We

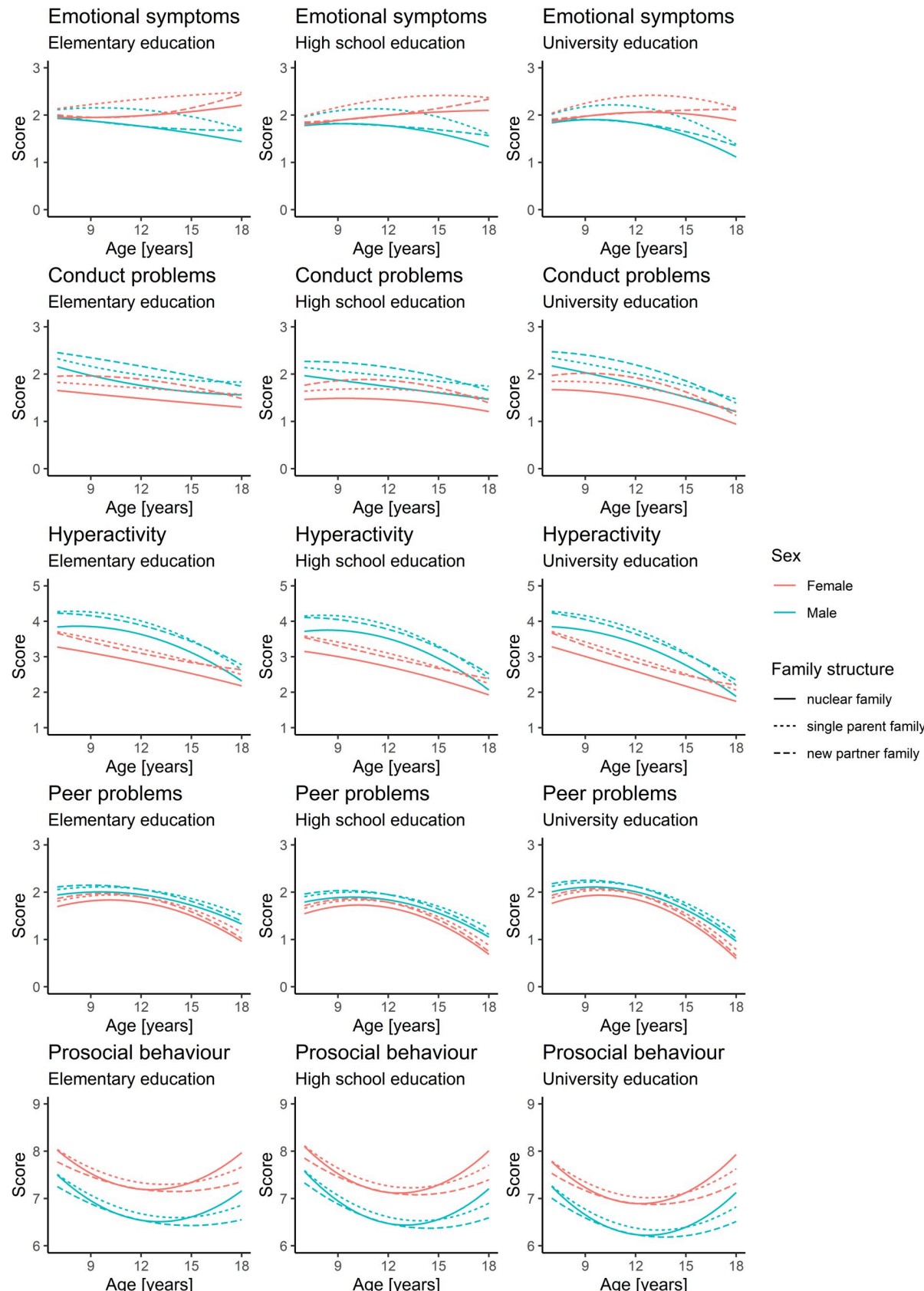

**Fig 1. Model 3 results.**

speculate that this difference may be explained by a private enterprise boom, especially among people with higher education. One or both parents embarking on a business career may have introduced a new measure of stress into the family environment which in return may have negatively influenced the children.

In general, the results of our analysis are in agreement with findings from western settings, indicating that higher education and nuclear family structure function as protective factors with respect to the psychosocial problems score. However, thanks to the unique setting, specific mechanics may work in a different way. For example, while the low income is generally associated with lower levels of education [1], this period for the Czech Republic is characterized by relatively low income discrepancy with regards to education. Household income is thus is determined rather by the number of household members with some form of financial income (work or social welfare) than by their level of education. Due to the fact, that we did not include income in our models, we speculate that the effect of poverty demonstrated in western settings [28,4] may be manifested mostly through the single parent family structure in our models and the socioeconomic status influences the child via a parent's education and work activities, but not through income.

In addition to the influence of maternal education and family structure at specific time-points, our longitudinal approach also mapped the overall trend during the course of a number of years. We believe that this approach offers better insight into relationships between variables and thus provides a more comprehensive image. The point of a longitudinal perspective is most apparent when differences between sexes are examined. While lower problem scores in females (except for emotional symptoms) are not an unexpected finding [29], differences in curve shapes between the sexes provide insight into children's psychosocial development. The effect of a child's sex on the overall shape of the curve is most apparent in the emotional symptoms and hyperactivity subscales. In the case of family structure, the effect on the problem score curve shape was minimal, and very similar findings could have been achieved using a cross-sectional approach. Only the prosocial behaviour score curve shape seems to be affected by family structure; the scores of children from nuclear families rise faster after 15y. On the other hand, in the case of all problem subscales, higher maternal education results in a steeper drop over time. We believe that this effect would be less clear or even completely hidden in case a cross-sectional approach were adopted.

Overall, the psychometric properties and relationships between subscales were comparable to those reported in other studies using SDQ [30]. This leads us to the conclusion that the issues with translation and scoring did not influence data quality in a serious manner. Possible limitations to our findings are primarily based on the fact that our data comes from a longitudinal study which suffers from a drop-out effect and is therefore prone to selection bias. The participants retained in the study have different characteristics that those who dropped out and it is quite difficult to estimate the magnitude of the effect due to a lack of information on subjects who dropped out. However, it has been shown for a study with a very similar design, that while selection bias leads to an underestimation of behaviour disorder incidence rates in a population, it does not bias the predictions and associations among variables [31]. Furthermore, our dataset suffers from missing data on important control variables including e.g. income. Another possible limitation is our use of maternal responses for the SDQ; while this enabled us to include more time-points, it also brings a possibility that the surveyed variables influence the mother's reporting of problems score, not the score itself. The last notable limitation is methodological; while mixed models provide a suitable framework for data with

repeated measurements and missing values, they may not be the best choice if the within-subject correlation structure does not meet the model's assumptions and the aim of the analysis is to provide predictions for individual subjects (which was not our primary aim). An alternative method may be the generalized estimating equations approach, which does not require the assumption regarding the correlation structure but has more strict assumptions about missing values [26].

Our findings show that associations between the children's psychosocial problems, socio-economic status and family structure in the Czech Republic are similar to associations reported in previous studies from western settings. Some minor differences may be explained by the specifics of the time period, but the overall direction of the results is very similar. The longitudinal approach to data proved to be useful and provided us with an important overview of the score over time.

In our further research, we aim to continue analysing data in a longitudinal manner, focusing on identified relationships between family structure and child's problems. In future analyses, we believe that it may be beneficial to pool the individual problem subscales into second-order internalising and externalising subscales, which may have better discriminant validity in population samples [32]. Looking more closely at family structure, one possible research direction is to explore the dynamics of its change, including e.g. the number of transitions and the direction of change. We also suggest differentiating and exploring individual factors such as family income, time spent with the child and extracurricular activities as well as comparing our analysis to similar longitudinal studies from western settings. We likewise propose a closer examination of family structure, especially as we believe that it would be beneficial to explore the support of extended family and quality of family relationships, which may have significant influence in single parent and new partner families.

## Acknowledgments

The authors of this study wish to thank the participating families as well as the gynaecologists, paediatricians, school heads and class teachers who took part. Our thanks also go to Dr. Lubomír Kukla, Ph.D., ELSPAC national coordinator 1990–2012, and the entire ELSPAC team.

## Author Contributions

**Conceptualization:** Daniela Kuruczova, Hynek Pikhart, Julie Bienertova-Vasku.

**Data curation:** Daniela Kuruczova.

**Formal analysis:** Daniela Kuruczova.

**Investigation:** Hynek Pikhart.

**Methodology:** Daniela Kuruczova, Jiri Jarkovsky.

**Resources:** Jana Klanova.

**Software:** Daniela Kuruczova, Jiri Jarkovsky.

**Supervision:** Jana Klanova, Hynek Pikhart, Julie Bienertova-Vasku.

**Visualization:** Daniela Kuruczova.

**Writing – original draft:** Daniela Kuruczova, Julie Bienertova-Vasku.

**Writing – review & editing:** Jana Klanova, Jiri Jarkovsky, Hynek Pikhart, Julie Bienertova-Vasku.

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
