## [Decision Letter · Decision Letter 0]

11 Feb 2020

PONE-D-19-25455

Socioeconomic characteristics, family structure and trajectories of children’s psychosocial problems in a period of social transition

PLOS ONE

Dear Ms. Kuruczová,

Thank you for submitting your manuscript to PLOS ONE. After careful consideration, we feel that it has merit but does not fully meet PLOS ONE’s publication criteria as it currently stands. Therefore, we invite you to submit a revised version of the manuscript that addresses the points raised during the review process.

We would appreciate receiving your revised manuscript by Mar 27 2020 11:59PM. To enhance the reproducibility of your results, we recommend that if applicable you deposit your laboratory protocols in protocols.io, where a protocol can be assigned its own identifier (DOI) such that it can be cited independently in the future. For instructions see: http://journals.plos.org/plosone/s/submission-guidelines#loc-laboratory-protocols

We look forward to receiving your revised manuscript.

Kind regards,

Orla Doyle

Academic Editor

PLOS ONE

Journal Requirements:

Reviewers' comments:

Reviewer's Responses to Questions

**Comments to the Author**

1. Is the manuscript technically sound, and do the data support the conclusions?

Reviewer #1: Yes

Reviewer #2: Yes

2. Has the statistical analysis been performed appropriately and rigorously? 

Reviewer #1: Yes

Reviewer #2: I Don't Know

3. Have the authors made all data underlying the findings in their manuscript fully available?

Reviewer #1: No

Reviewer #2: Yes

4. Is the manuscript presented in an intelligible fashion and written in standard English?

Reviewer #1: Yes

Reviewer #2: Yes

5. Review Comments to the Author

Reviewer #1: General Comments:

The paper provides an analysis of the relationship between family structure, SES and children’s mental health within a unique setting, and from a longitudinal perspective which represents a growing literature in Western settings. The length of follow up from middle childhood through adolescence, and gender comparisons are a strength of the study with child gender differences rarely considered. A limitation is the lack of control and possible explanatory variables such as income.

The analysis is sound although some clarity is needed in the methods and results sections. The conclusions drawn are appropriate.

The authors have not included a Data Availability Statement in their manuscript although I assume that the ELSPAC data is publicly available to researchers with ethics approval. This will need to be submitted for final consideration of the manuscript

The manuscript is well written but importantly needs reviewing for minor grammatical errors throughout and to conform completely with standard English writing (e.g. “a child’s problems” – the “a” has been omitted).

Specific Comments:

Abstract is fine apart from some minor grammatical fixes (e.g. “Data from the Czech part of the European Longitudinal Study of Pregnancy and Childhood..”).

Introduction

Although some review papers are referenced, there are further studies with longitudinal design in addition to Flouri that could be referenced e.g Pearce and colleagues (2014) using data from the UK Millenium Cohort study; Ryan & Claessens (2013) Magnuson & Berger (2009), Fomby & Cherlin (2007) using the NLSY; and Perales et al (2015) using data from the Longitudinal Study of Australian Children. Based on these studies, more could be said about the timing and nature of family transitions (in particular by of the child which is a focus of this paper).

On page 3 (line 56) the authors state that “the role of family structure is more straightforward in single-parent families..” Please expand to explain what this means. On Line 61, what does the “possible non-linear” relationship look like.

Although the study provides a unique setting on a post-communist period, do the authors have any views about whether they expect to similar associations in the Czech Republic, exacerbated due to the rate of change or perhaps with less impact on children due to greater individual freedom and choice?

Materials and Methods

Study population – The sample loss over time is large. It would be informative to summarise attrition bias over time if possible/available in other methodology documents?

Please report how missing item response was dealt with in scoring the SDQ.

Family structure – what happened with families where the child lived with the biological father but not the biological mother (either in a new partner family or as a lone father family). I assume they excluded? If so, what was the sample size after these exclusions?

Socioeconomic status. Although household income correlates with education, I wonder if it is too strong to exclude income from the analysis which is an important (explanatory) factor in the relationship between sole parent families particularly and child outcomes.

Cronbach’s alpha for SDQ scale is probably better reported in the Methods section than in the results.

Results.

P11. Line 169. Make a statement in respect to family structure and age interactions i.e. there were none and therefore the associations did not vary by the follow-up age. Is that the correct interpretation?

The key for Figures 1 & 2 does not work well in black and white. Gender will need to be distinguished by colour.

Discussion.

As per the comment above, what is the conclusion regarding timing/child age?

P13. Line 223-26. Please clarify comment (and perhaps re-word sentences) about the relationship between income and education and whether sole parent effects are thought to be partly explained by income/poverty or not in this setting. If there is low income discrepancy in regards to education during this period, then is that justification for including in the models along with maternal education?

P13. Lines 229-234. The gender differences in problems is well known and of less interest than whether there was a gender difference in the relationship between family structure and psychosocial problems over time, which there doesn’t seem to be.

Study limitations are appropriately described.The lack of control variables representing possible selection effects is another limitation.

Reviewer #2: The manuscripts presents longitudinal analyses on predictors of change in child wellbeing from age 7 to 18. It’s main interest is on family structure as a determinant of psychosocial problems over time. It also includes a social inequality perspective by taking mother’s education into account. Using data from different time points is a major strength of the analyses as it reveals distinct patterns of the development of psychosocial problems as children grow older. On the other hand there are several points which need clarification. I list them as they appear in the text.

Abstract

The abstract misses:

1. The number of observations (at least for baseline)

2. The number of measurement points

3. What is a ‘western setting’? This term is unclear to me

4. A justification for the last sentence. The advantages of longitudinal data are not mentioned in the results section

Introduction

A general point is the missing theoretical foundation. The introduction is rather short and the rationale for the investigation does not become clear to me. It is mentioned that associations are already studied in “western setting” (whatever this means) but no details are provided. What kind of analyses is missing so far? Implicitly I understand that data from Czech should be compared with other countries but what is the hypothesis you want to study? Why should we expect any differences in the association between child health and family structure between Czech and other countries?

Next: the complex association between family structure and SES is not described and it remains unclear why both characteristics are investigated together in this study.

To conclude: the theory/rationale of the study should be elaborated and more details from previous research are needed.

Some further minor points:

Line 48: You mention “Multiple studies” but not a single study is cited.

Line 43 and line 50: You mention differences in income between high and low SES. As income is a core indictor for SES this sentence does not make sense.

Line 58: it is said that most studies use cross-sectional designs but there are not references for this statement

Line 61: a non-linear relationship is mentioned but of what kind?

Methods

Please provide information on the sampling. What was the baseline response? Where there any kind of measures to minimize drop-off?

The justification to use the mother’s education at birth is convincing but nonetheless this indicator has its risks. Especially younger mothers may gain additional educational degrees after child birth. Did you considered sensitivity analyses with other SES indicators or with a measurement of education at the last measurement point?

I am not familiar with the SDQ instrument. Is it an international standard for measuring problems in children? Why is the scale peer problems part of an internalizing scale (social relations seem to be external by definition so me).

Results

Drop-off rates are high. It would be interesting to know if the rates were different in different family structures and educational groups in order to assess a possible bias.

Statistics are advanced and impressive but I have to admit not to be an expert in this kind of modelling. It would be helpful to get a bit more explanation about what the estimators mean in the text of the session.

6. PLOS authors have the option to publish the peer review history of their article (what does this mean?). If published, this will include your full peer review and any attached files.

Reviewer #1: No

Reviewer #2: No

---

## [Author Response · Author response to Decision Letter 0]

27 Mar 2020

Reviewer #1: General Comments:

The paper provides an analysis of the relationship between family structure, SES and children’s mental health within a unique setting, and from a longitudinal perspective which represents a growing literature in Western settings. The length of follow up from middle childhood through adolescence, and gender comparisons are a strength of the study with child gender differences rarely considered. A limitation is the lack of control and possible explanatory variables such as income.

The analysis is sound although some clarity is needed in the methods and results sections. The conclusions drawn are appropriate.

The authors have not included a Data Availability Statement in their manuscript although I assume that the ELSPAC data is publicly available to researchers with ethics approval. This will need to be submitted for final consideration of the manuscript.

• The data availability statement has been added to the manuscript. It was originally wrongly omitted, we apologize for the confusion.

The manuscript is well written but importantly needs reviewing for minor grammatical errors throughout and to conform completely with standard English writing (e.g. “a child’s problems” – the “a” has been omitted).

• The manuscript was proof-read by the native speaker of English with sufficient expertise in scientific English.

Specific Comments:

Abstract is fine apart from some minor grammatical fixes (e.g. “Data from the Czech part of the European Longitudinal Study of Pregnancy and Childhood..”).

• The typos were corrected, we apologize for this. 

Introduction

Although some review papers are referenced, there are further studies with longitudinal design in addition to Flouri that could be referenced e.g Pearce and colleagues (2014) using data from the UK Millenium Cohort study; Ryan & Claessens (2013) Magnuson & Berger (2009), Fomby & Cherlin (2007) using the NLSY; and Perales et al (2015) using data from the Longitudinal Study of Australian Children. Based on these studies, more could be said about the timing and nature of family transitions (in particular by of the child which is a focus of this paper).

• Initially, we focused on the papers that used SDQ as a diagnostic tool, which narrowed down the number of studies mentioned in the Introduction part. However, this decision might have been too restrictive as the suggested studies bring more insight into family structure dynamics. The Introduction was completely rewritten to include more relevant studies, including several suggested ones.

On page 3 (line 56) the authors state that “the role of family structure is more straightforward in single-parent families..” Please expand to explain what this means. On Line 61, what does the “possible non-linear” relationship look like.

• Both sections have been rewritten to be more explicit and more descriptive.

Although the study provides a unique setting on a post-communist period, do the authors have any views about whether they expect to similar associations in the Czech Republic, exacerbated due to the rate of change or perhaps with less impact on children due to greater individual freedom and choice?

• The section on the author’s expectations was added. In general, we expected that the negative impact of low maternal education and single-parent would be present in our sample, but the effect size was expected to be smaller.

Materials and Methods

Study population – The sample loss over time is large. It would be informative to summarise attrition bias over time if possible/available in other methodology documents?

• Unfortunately, no official summary of attrition bias for ELSPAC study is available. We created a summary on attrition bias specifically for our article and added it to the sample characteristics in the Results section.

Please report how missing item response was dealt with in scoring the SDQ.

• Information on how missing items were approached was added to the manuscript.

Family structure – what happened with families where the child lived with the biological father but not the biological mother (either in a new partner family or as a lone father family). I assume they excluded? If so, what was the sample size after these exclusions?

• Information on other family structures was added to the text. As is correctly assumed, families, where the child lived with the father, were excluded. These exclusions did not significantly affect the sample size, as the number of families, where the child stayed with the father, was less than 10.

Socioeconomic status. Although household income correlates with education, I wonder if it is too strong to exclude income from the analysis which is an important (explanatory) factor in the relationship between sole parent families particularly and child outcomes.

• We agree that the income should be included in the analysis, and we do have equalized family income variable available. However, this variable significantly increases the number of missing answers and amplifies the selection bias as it is missing mostly for lower educated mothers. We fitted the models also with income included and found out that the results were mostly the same. Some significant interactions emerged, but mostly concerned the least numerous “new partner family” category and seemed rather spurious. The decision to exclude the income was not an easy one, but we believe that the presented results are more robust and less biased.

Cronbach’s alpha for SDQ scale is probably better reported in the Methods section than in the results.

• Thank you for this suggestion; the information was moved to the methods section.

Results.

P11. Line 169. Make a statement in respect to family structure and age interactions i.e. there were none and therefore the associations did not vary by the follow-up age. Is that the correct interpretation?

• In Model 2, we found no significant interactions between family structure and age. However, we are using the nuclear family as the reference value, and therefore the coefficients Age and Age2 (which are both significant for almost all problem subscales) show that the relationship between age and the problem score is a quadratic curve in nuclear families. The lack of significant interactions between age and other family structures shows that there is no significant difference in the shape of the quadratic curve. We added more detailed info on interpretation of the models in the statistical analysis section in order to make understanding and interpreting our results easier for the reader.

The key for Figures 1 & 2 does not work well in black and white. Gender will need to be distinguished by colour.

• Thank you for this feedback, the figures were re-created in colour.

Discussion.

As per the comment above, what is the conclusion regarding timing/child age?

• As illustrated in Figure 1, the problem scores are decreasing in time for all subscales, with one notable exception – emotional problems in women. The prosocial behaviour curve has a U-shape with a minimum of approximately 13y. This summary was added to the beginning of the discussion; thank you for pointing out that it is missing.

P13. Line 223-26. Please clarify comment (and perhaps re-word sentences) about the relationship between income and education and whether sole parent effects are thought to be partly explained by income/poverty or not in this setting. If there is low income discrepancy in regards to education during this period, then is that justification for including in the models along with maternal education?

• The section was rewritten to be more explicit. We agree that including income would be theoretically the best solution, but we outline our reasons for not doing so in the Materials and Methods section above.

P13. Lines 229-234. The gender differences in problems is well known and of less interest than whether there was a gender difference in the relationship between family structure and psychosocial problems over time, which there doesn’t seem to be.

• We agree with this point, the section was modified to reflect this.

Study limitations are appropriately described.The lack of control variables representing possible selection effects is another limitation.

• A lack of control variables was added to the limitations list.

Reviewer #2: The manuscripts presents longitudinal analyses on predictors of change in child wellbeing from age 7 to 18. It’s main interest is on family structure as a determinant of psychosocial problems over time. It also includes a social inequality perspective by taking mother’s education into account. Using data from different time points is a major strength of the analyses as it reveals distinct patterns of the development of psychosocial problems as children grow older. On the other hand there are several points which need clarification. I list them as they appear in the text.

Abstract

The abstract misses:

1. The number of observations (at least for baseline)

2. The number of measurement points

3. What is a ‘western setting’? This term is unclear to me

4. A justification for the last sentence. The advantages of longitudinal data are not mentioned in the results section

• Thank you for pointing out these shortcomings; information was added to the abstract.

• We used the western setting to distinguish countries with no period of the communist regime. We edited the Introduction part to make this distinction more clear.

• The advantages of the longitudinal approach are not mentioned in the Results section; they are discussed in the Discussion part of the paper.

Introduction

A general point is the missing theoretical foundation. The introduction is rather short and the rationale for the investigation does not become clear to me. It is mentioned that associations are already studied in “western setting” (whatever this means) but no details are provided. What kind of analyses is missing so far? Implicitly I understand that data from Czech should be compared with other countries but what is the hypothesis you want to study? Why should we expect any differences in the association between child health and family structure between Czech and other countries?

Next: the complex association between family structure and SES is not described and it remains unclear why both characteristics are investigated together in this study.

To conclude: the theory/rationale of the study should be elaborated and more details from previous research are needed.

• The introduction was completely rewritten to reflect the rationale for our study and also order to explain the rationale for comparing the Czech sample with other countries.

Some further minor points:

Line 48: You mention “Multiple studies” but not a single study is cited.

• We used this sentence as introductory for the whole paragraph; the individual studies are subsequently described and cited in the following sentences.

Line 43 and line 50: You mention differences in income between high and low SES. As income is a core indictor for SES this sentence does not make sense.

• We apologize for the somewhat unfortunate initial wording; we certainly consider income one of the indicators for SES. The text was edited to reflect this more clearly.

Line 58: it is said that most studies use cross-sectional designs but there are not references for this statement

• Again, we consider this an introductory sentence for the whole paragraph.

Line 61: a non-linear relationship is mentioned but of what kind?

• We expanded this section to explain the nature of the non-linear relationship more clearly.

Methods

Please provide information on the sampling. What was the baseline response? Where there any kind of measures to minimize drop-off?

• The general information on sampling methodology for this cohort can be found in the article by Piler et al. [20], we intentionally do not include this information in our article.

The justification to use the mother’s education at birth is convincing but nonetheless this indicator has its risks. Especially younger mothers may gain additional educational degrees after child birth. Did you considered sensitivity analyses with other SES indicators or with a measurement of education at the last measurement point?

• Yes, we re-run the analysis using the variable “highest known maternal education.” The results and conclusions remained the same. We decided to use the “maternal education at birth” variable because it is available for the majority of subjects. We also believe that the “highest known education” brings unpredictable bias for mothers that dropped out during the study.

I am not familiar with the SDQ instrument. Is it an international standard for measuring problems in children? Why is the scale peer problems part of an internalizing scale (social relations seem to be external by definition so me).

• The instrument SDQ is commonly used to screen for children’s psychosocial problems and was translated to over 80 languages. Multiple large-scale longitudinal studies use SDQ, e.g. ALSPAC [21], Millennium Cohort Study [5] or KIDSCREEN [13].

• The peer problems subscale is considered internalizing (as per guidelines by authors of SDQ), because it focuses on child’s internal perception of their relationship with peers. Both externalizing problem scales (conduct and hyperactivity) focus on behavioural manifestations of these problems [11].

Results

Drop-off rates are high. It would be interesting to know if the rates were different in different family structures and educational groups in order to assess a possible bias.

• More information on assessing possible bias and its directions were added in the Results part.

Statistics are advanced and impressive but I have to admit not to be an expert in this kind of modelling. It would be helpful to get a bit more explanation about what the estimators mean in the text of the session.

• We were unsure whether to include this information or not, thank you for suggesting that it might be better to include it. The information on the model and its interpretation was added to the Methods part.

---

## [Decision Letter · Decision Letter 1]

9 Apr 2020

PONE-D-19-25455R1

Socioeconomic characteristics, family structure and trajectories of children’s psychosocial problems in a period of social transition

PLOS ONE

Dear Ms. Kuruczová,

Thank you for submitting your manuscript to PLOS ONE. Thank you for revising your paper. One of the reviewers has made some suggestions of very minor changes. Therefore, we invite you to submit a revised version of the manuscript that addresses the points raised.

We would appreciate receiving your revised manuscript by May 24 2020 11:59PM. To enhance the reproducibility of your results, we recommend that if applicable you deposit your laboratory protocols in protocols.io, where a protocol can be assigned its own identifier (DOI) such that it can be cited independently in the future. For instructions see: http://journals.plos.org/plosone/s/submission-guidelines#loc-laboratory-protocols

We look forward to receiving your revised manuscript.

Kind regards,

Orla Doyle

Academic Editor

PLOS ONE

Reviewers' comments:

Reviewer's Responses to Questions

**Comments to the Author**

1. If the authors have adequately addressed your comments raised in a previous round of review and you feel that this manuscript is now acceptable for publication, you may indicate that here to bypass the “Comments to the Author” section, enter your conflict of interest statement in the “Confidential to Editor” section, and submit your "Accept" recommendation.

Reviewer #1: (No Response)

Reviewer #2: All comments have been addressed

2. Is the manuscript technically sound, and do the data support the conclusions?

Reviewer #1: Yes

Reviewer #2: Yes

3. Has the statistical analysis been performed appropriately and rigorously? 

Reviewer #1: Yes

Reviewer #2: Yes

4. Have the authors made all data underlying the findings in their manuscript fully available?

Reviewer #1: Yes

Reviewer #2: (No Response)

5. Is the manuscript presented in an intelligible fashion and written in standard English?

Reviewer #1: Yes

Reviewer #2: Yes

6. Review Comments to the Author

Reviewer #1: The authors have undertaken substantial revision and done an excellent job in responding to the reviewer’s comments and as such, the manuscript is greatly improved. I have some further minor suggestions as below. The manuscript needs a second thorough review for grammatical errors – I have flagged some of these.

Abstract

Grammatical errors.

“Findings from western setting(s)…” and throughout the manuscript.

“Modelled as (a) quadratic growth curve”

“Essential for (a) more complex view”

Introduction

Line 42. “a decrease in (a) child’s psychosocial problems” (and Line 53)

Consider some rewording in the following places:

Line 48-49. Not all re-partnered families are married couples. There are plenty of studies that separate re-partnered from nuclear families – the focus on single parent families in family structure has shifted and multiple categories (including step and blended families) are standard in statistical collections.

Line 55. Explain the practical interpretation of the Flouri study for those not familiar with quadratic growth curves i.e. it’s not immediately obvious what a vertical shift of the growth curve means in respect to family structure and SES.

Line 69. Did the Kidscreen study [13] include family structure and or SES which would make this a more appropriate point to make – the effect of something like parental mental health is universal? You have noted lower SES in Eastern European countries but how did the mental health of Czech children, and the relationship of MH with SES compare with other countries?

Line 87. Why do you expect a smaller effect size? Suggest adding a because.. statement to the end of this sentence.

Materials and methods

Looks good.

The analytical sample varies across Models with the addition of family structure and education variables. Although the analytical sample is defined by SDQ score, please clarify if there are cases in the 3,261 that have no data on predictive variables at any follow up? That is, if I look at Table 2, the Ns are relevant for Model 1 but what about Models 2 & 3 when family structure and maternal education are added?

Results

Line 196. Correction – “Unfortunately, from this point on the information on the non-analytic sample is limited”.

Table 1. It doesn’t make much sense to have data about family structure for the non-analytic sample beyond that measured at birth. Suggest removing these figures from the table.

In terms of the analytical sample, I wonder if reporting the % of those in different family structures of the non- missing cases would be more informative i.e. keep the N for Missing but exclude from the denominator. Then we can easily see the change in family structure based on available cases albeit with likely biased attrition in the sample. You could also add the year to the row labels e.g. “Family structure at 7y (1998/99) to connect back to the period in time which is interesting given the dramatic change.

In terms of assessing differential attrition in family types, this could be assessed based on drop outs from year 7 e.g. proportion of original, single and re-partnered families still in the sample at year 11 etc.. Because you have already excluded those with no SDQ data, this may not show a lot more.

Discussion

Line 258. Are you able to report here or perhaps in the intro the increase in the proportion of single parent and re-partnered families during this time based on other sources?

Line 259. Be careful in this paragraph not to make it seem that all single parent families are burdened. Many children in single parent families also do well. (delete The) Single parenthood comes with.. “

Line 267. Reference [8] – was it the case in this study that internalizing problems were not connected to quality of relationship with the father?

Line 270 “For (the) hyperactivity subscale..” Please explain this sentence in practical terms.. “mostly provided significant negative vertical shift, meaning that…” hyperactivity problems were lower in children of university educated mothers across ages”?

Line 277. “higher socioeconomic status is a protective factor for prosocial behaviour” remove ‘the’

Line 283 “lower income is usually associated also with lower education” remove ‘the’

Line 292. “Provides (a) more comprehensive picture”

Line 296 Is it worth mentioning family structure and prosocial behaviour over time (less improvement in single parent and new partner families based on significant difference in curve?

Line 303. “Comes from (a) longitudinal study..”

Line 313. Does your model meet assumptions? Mentioning this here is a concern unless you can report that it does meet assumptions in the Methods.

Line 320. Further research – perhaps consider the total SDQ scale and internalising/externalising scales which tend to be more reliable than individual sub-scales in population samples (e.g. see Goodman et al., 2010).

Reviewer #2: (No Response)

7. PLOS authors have the option to publish the peer review history of their article (what does this mean?). If published, this will include your full peer review and any attached files.

Reviewer #1: No

Reviewer #2: No

---

## [Author Response · Author response to Decision Letter 1]

15 May 2020

Dear Sir or Madam,

We would like to thank you for your feedback. We appreciate the detailed response and suggestions for improving our paper. We attempted to incorporate most of the comments and believe that the manuscript quality improved. To ensure proper use of English, the text was revised by a professional editor. 

Please find our answers to the individual comments in the "Response to Reviewers" file.

Best regards,

Daniela Kuruczová

---

## [Editor Report · Decision Letter 2]

19 May 2020

Socioeconomic characteristics, family structure and trajectories of children’s psychosocial problems in a period of social transition

PONE-D-19-25455R2

Dear Dr. Kuruczová,

We are pleased to inform you that your manuscript has been judged scientifically suitable for publication and will be formally accepted for publication once it complies with all outstanding technical requirements.

With kind regards,

Orla Doyle

Academic Editor

PLOS ONE
---

## [Editor Report · Acceptance letter]

28 May 2020

PONE-D-19-25455R2 

Socioeconomic characteristics, family structure and trajectories of children’s psychosocial problems in a period of social transition 

Dear Dr. Kuruczova:

I am pleased to inform you that your manuscript has been deemed suitable for publication in PLOS ONE. Congratulations! Your manuscript is now with our production department. 

With kind regards,

on behalf of

Dr. Orla Doyle 

Academic Editor

PLOS ONE